# Optical source apportionment and radiative effect of light-absorbing carbonaceous aerosols in a tropical marine monsoon climate zone: The importance of ship emissions

Qiyuan Wang[1,2], Huikun Liu[1], Ping Wang[3], Wenting Dai[1], Ting Zhang[1], Youzhi Zhao[3], Jie Tian[1], Wenyan Zhang[1], Yongming Han[1,2], Junji Cao[1,2]

[1]Key Laboratory of Aerosol Chemistry and Physics, State Key Laboratory of Loess and Quaternary Geology, Institute of Earth Environment, Chinese Academy of Sciences, Xi'an 710061, China
[2]CAS Center for Excellence in Quaternary Science and Global Change, Xi'an 710061, China
[3]Hainan Tropical Ocean University, Sanya 572022, China

*Correspondence to*: Qiyuan Wang (wangqy@ieecas.cn) and Junji Cao (cao@loess.llqg.ac.cn)

**Abstract.** Source-specific optical properties of light-absorbing carbonaceous (LAC) aerosols in the atmosphere are poorly understood because they are generated by various sources. In this study, a receptor model combining multi-wavelength absorption and chemical species was used to explore the source-specific optical properties of LAC aerosols in a tropical marine monsoon climate zone. The results showed that biomass burning had the largest contribution to average LAC absorption. However, ship emissions emerged as the dominant contributors (44–45%) when the air masses originated from the South China Sea. Additionally, the source-specific Absorption Ångström Exponent (AAE) indicated that black carbon (BC) was the dominant LAC aerosol in ship and motor vehicle emissions. Moreover, brown carbon (BrC) was present in biomass-burning emissions. The source-specific mass absorption cross-section (MAC) showed that BC from ship emissions had a stronger light-absorbing capacity compared to emissions from biomass burning and motor vehicles. The BrC MAC derived from biomass burning was also smaller than the BC MAC and was highly depended on wavelength. Furthermore, radiative effect assessment indicated a comparable atmospheric forcing and heating capacity of LAC aerosols between biomass burning and ship emissions. This study provides insights into the optical properties of LAC aerosols from various sources. It also sheds more light on the radiative effects of LAC aerosols generated by ship emissions.

# 1 Introduction

Carbonaceous aerosols are abundant in $PM_{2.5}$ (particulate matter with an aerodynamic diameter $\leq 2.5$ μm) (e.g., 20–50% of $PM_{2.5}$ mass, Putaud et al., 2010; Tao et al., 2017) and have extensively been explored due to their implications on global climate forcing (IPCC, 2013). Among the complex carbonaceous compounds are the light-absorbing carbonaceous (LAC) aerosols which are mainly associated with absorption of light. LAC aerosols consist of black carbon (BC) and brown carbon (BrC). BC is a short-lived climate forcer with a strong ability to absorb sunlight. Moreover, it is regarded as the second largest contributor of positive anthropogenic climate forcing after carbon dioxide (Bond et al., 2013). On the other hand, BrC refers to a class of light-absorbing organic compounds with enhanced light absorption at short wavelengths (e.g., near-ultraviolet region). Therefore, it is a potential contributor to atmospheric heating at both global and regional scales (Laskin et al., 2015).

The optical properties of LAC aerosols are closely related to their sources as well as atmospheric conditions and secondary processing. However, distinguishing source-specific light absorption by LAC from a mixture of aerosols in the atmosphere is still a challenge. It is possible to use multi-wavelength light absorption data to identify optical source apportionment based on the Beer-Lambert's Law (e.g., aethalometer model and multi-wavelength absorption analyzer model, Sandradewi et al., 2008; Massabò et al., 2015), which can typically explain two different types of sources (e.g., fossil fuels versus biomass burning). Results from this method are highly dependent on the use of the source-specific Absorption Ångström Exponent (AAE). However, due to lack of source-specific AAE data, most studies use empirical values reported in previous literature (e.g., Healy et al., 2017; Küpper et al., 2018; Zheng et al., 2019). This may create inconsistencies in the reported results because the source-specific AAE varies with the type of fuels and their burning efficiencies (Tian et al., 2019).

In addition, optical source apportionment can be obtained using receptor models (e.g., Positive Matrix Factorization (PMF) and Multilinear Engine (ME2)). Several studies have utilized receptor models to identify sources, first based on the sole chemical species or mass spectra information. Thereafter, a multiple linear regression model is used to apportion the contribution of each source to the optical parameters of an aerosol (Qin et al., 2018; Tian et al., 2020). This method may be referred to as indirect optical source apportionment. In contrast, Forello et al. (2019) coupled chemical species with multi-

wavelength absorption in ME2 to directly perform optical source apportionment. Compared to the indirect approach, the additional optical data in receptor models can improve the performance of source apportionment because each source has its own optical features. Furthermore, it may eliminate potential uncertainties caused by multiple operations in the indirect approach. However, the application of direct optical source apportionment is scarce at the moment.

Alternatively, laboratory studies may effectively be used to explore the optical properties of LAC from a specific source (e.g., vehicle engine exhaust, coal combustion, and biomass burning) (Tian et al., 2019; Xie et al., 2017). However, the optical properties of LAC may significantly change due to the complex atmospheric processes that they undergo after emission into the atmosphere. Therefore, it is critical to identify LAC aerosols from different sources in the atmosphere using specific methods in order to obtain their optical properties. Furthermore, to the best of our knowledge, there is no study focusing on the optical properties of ship exhaust-related LAC aerosols in the atmosphere. This presents a challenge to our understanding of the role of ship emissions on the climate considering that it is a significant part of discharge from the transport sector.

In this study, multi-wavelength aerosol light absorption and chemical species were measured in Sanya, a coastal city in China. This was done to investigate the optical properties of LAC aerosols from ship emissions and other sources. A dataset combining optical and chemically speciated data was used simultaneously in a receptor model to obtain the optical source apportionment. Afterwards, the source-specific optical properties of LAC aerosols were determined and characterized. Finally, the impact of radiative effect induced by LAC aerosols from different sources was evaluated. This study provides insights into the source-specific optical properties of LAC aerosols from various sources. Additionally, it reinforces knowledge on the radiative effects of LAC aerosols.

## 2 Methodology

### 2.1 Sampling site

The sampling site is located in Sanya, a small city (an area of 1921.5 $km^2$ and a total population of 0.59 million as at 2017) in the southernmost tip of the Hainan Island in Southern China (Fig. S1).

Comprehensive measurements were taken in spring from 12<sup>th</sup> April to 14<sup>th</sup> May 2017 on the rooftop of a teaching building (about 20 m above ground level) in Hainan Tropical Ocean University (18.30° N, 109.52° E). The sampling site is predominantly an educational and residential area with typical urban sources of emission including vehicles and cooking appliances. Sanya lies within a tropical marine monsoon climate zone therefore the weather was warm (temperature $= 28 \pm 3°C$) and wet (relative humidity $= 81 \pm 12\%$) during the study.

## 2.2 Online and offline measurements

A model AE33 aethalometer (Magee Scientific, Berkeley, CA, USA) was used to determine the light absorption coefficients of the aerosols at multi-wavelengths (Abs($\lambda$), $\lambda$ is wavelength) with a $PM_{2.5}$ cyclone (SCC 1.829, BGI Inc. USA). Briefly, the collected particles were desiccated using a Nafion® dryer (MD-700-24S-3; Perma Pure, Inc., Lakewood, NJ, USA) before measurement with the AE33 aethalometer. As shown in Fig. S2, the loss of Abs($\lambda$) caused by the dryer was ignored. Afterwards, seven light emitting diodes ($\lambda = 370, 470, 520, 590, 660, 880,$ and 950 nm) in the AE33 aethalometer were used to irradiate the filter deposition spot to obtain light attenuation as previously described (Drinovec et al., 2015). Since the AE33 aethalometer records the BC mass concentrations, the Abs($\lambda$) at each wavelength were retrieved by getting the product of BC mass concentration ([BC]) and mass absorption cross-section (MAC) used in the instrument (Abs($\lambda$) = [BC] × MAC) (Drinovec et al., 2015). One of the advantages of AE33 aethalometer is that it resolves the filter loading effect using a dual-spot compensation technique. Further details regarding the principles of operation of the AE33 aethalometer have been outlined by Drinovec et al. (2015).

In addition, a Photoacoustic Extinctiometer (PAX, Droplet Measurement Technologies, Boulder, CO, USA) was used to directly measure aerosol light absorption at $\lambda = 532$ nm. It was set in parallel with the AE33 aethalometer using the same $PM_{2.5}$ cyclone and Nafion® dryer. Briefly, the PAX adopts an intracavity photoacoustic technique, with a modulated laser beam heating up the sampled particles in an acoustic chamber. The pressure wave generated from heating is then detected by a sensitive microphone. Moreover, aerosol light scattering can be measured using a wide-angle integrating reciprocal nephelometer in a scattering chamber. In this study, different concentration gradients of ammonium

sulphate and freshly-generated propane soot were used to calibrate light scattering and absorption measurements, respectively. The calibration procedure was described in detail by Q. Wang et al. (2018a). The PM$_{2.5}$ quartz-fiber filters (8 × 10 inch) (QM/A; GE Healthcare, Chicago, IL, USA) were collected during the day (from 08:00 to 20:00) and at night (from 20:00 to 08:00 the next day) using a high-volume air sampler (Tisch Environmental, Inc., USA) with a flowrate of 1.13 m$^3$ min$^{-1}$. Before sampling, the blank quartz-fiber filters were heated in a muffle furnace at 805 °C for 3h to remove possible impurities. After sampling, the quartz-fiber filters were saved in a freezer at about -20 °C to minimize evaporation of volatile material before chemical analyses. Finally, field blanks were collected and analysed to eliminate potential background artifacts.

The collected quartz-fiber filters were used to analyse inorganic elements, carbonaceous matter, water-soluble ions, and organics. An Energy-dispersive X-ray Fluorescence (ED-XRF) spectrometry (Epsilon 5 ED-XRF, PANalytical B.V., Netherlands) was used to determine the Titanium (Ti), Vanadium (V), Manganese (Mn), Ferrum (Fe), Nickel (Ni), Copper (Cu), Zinc (Zn) and Bromine (Br) quantities. A detailed description of the principles of ED-XRF has been highlighted by Xu et al. (2012). Moreover, a thermal/optical carbon analyzer (Desert Research Institute Model 2001, Atmoslytic Inc., Calabasa, CA, USA) was used to analyse organic carbon (OC) and elemental carbon (EC). A detailed analytical procedure has been described elsewhere (Chow et al., 2007). An Ion Chromatograph (IC, Dionex 600; Dionex Corporation, Sunnyvale, CA, USA) was also used to quantify the water-soluble cations (i.e., Na$^+$, K$^+$, Mg$^{2+}$, Ca$^{2+}$, and NH$_4^+$) and anions (i.e., Cl$^-$, NO$_3^-$, and SO$_4^{2-}$) as described by Zhang et al. (2011). Finally, an in-injection port Thermal Desorption (TD) coupled with an Agilent 7890/5975C Gas Chromatography/Mass Spectrometer (GC/MS) (Agilent Technologies, Santa Clara, CA, USA) was used to determine the hopanes using a protocol described by J. Wang et al. (2016).

## 2.3 Segregation of BC and BrC absorption

The Abs($\lambda$) consisted of light absorption from both LAC aerosols (BC and BrC) and mineral dust (Wang et al., 2013). Therefore, LAC absorption (Abs$_{LAC}$($\lambda$)) was calculated as follows:

$$Abs_{LAC}(\lambda) = Abs_{BC}(\lambda) + Abs_{BrC}(\lambda) = Abs(\lambda) - Abs_{mineral}(\lambda) \tag{1}$$

where $Abs_{BC}(\lambda)$, $Abs_{BrC}(\lambda)$, and $Abs_{mineral}(\lambda)$ were absorption of light by BC, BrC, and mineral dust at $\lambda$ = 370, 470, 520, 590, 660 or 880 nm, respectively (in unit of $Mm^{-1}$). The $Abs_{mineral}(\lambda)$ was retrieved from the optical source apportionment as discussed in section 3.2. With an assumption of BC only absorbing at $\lambda$ = 880 nm, the $Abs_{BC}(\lambda)$ at wavelengths of 370, 470, 520, 590, and 660 was extrapolated as follows:

$$Abs_{BC}(\lambda) = Abs(880) \times \left(\frac{\lambda}{880}\right)^{-AAE_{BC}} \qquad (2)$$

where $AAE_{BC}$ represents BC AAE, which was assumed to be 1.1 based on a study by Lack and Langridge (2013). Combining Eqs. (1) and (2) gave the following equation:

$$Abs_{BrC}(\lambda) = Abs(\lambda) - Abs(880) \times \left(\frac{\lambda}{880}\right)^{-AAE_{BC}} - Abs_{mineral}(\lambda) \qquad (3)$$

From the perspective of emission and formation, the $Abs(\lambda)$ could be divided into light absorption contributed by primary emissions ($Abs_{pri}(\lambda)$) and secondary formation ($Abs_{sec}(\lambda)$). Therefore, the $Abs(\lambda)$ could be calculated as follows:

$$Abs(\lambda) = Abs_{pri}(\lambda) + Abs_{sec}(\lambda) \qquad (4)$$

A BC-tracer method was utilized to separate $Abs_{sec}(\lambda)$ from $Abs_{pri}(\lambda)$ and the Eq. (4) could further be developed as follows (Wang et al., 2019a):

$$Abs_{sec}(\lambda) = Abs(\lambda) - \left(\frac{Abs(\lambda)}{BC}\right)_{pri} \times [BC] \qquad (5)$$

where $\left(\frac{Abs(\lambda)}{BC}\right)_{pri}$ described the ratio of $Abs(\lambda)$ to BC mass concentration in primary emissions (in unit of $m^2\ g^{-1}$) and [BC] denoted the mass concentration of BC in the atmosphere (in unit of $\mu g\ m^{-3}$). This was retrieved from the relationship between $Abs(880)$ measured by the AE33 aethalometer and EC mass concentration. Finally, the $\left(\frac{Abs(\lambda)}{BC}\right)_{pri}$ ratio was determined using a minimum $R$-squared (MRS) method previously described by Wang et al. (2019a).

## 2.4 Estimation of optical parameters

AAE reflects spectral dependence of aerosol light absorption and can be used to distinguish the chemical composition of LAC aerosols. For example, LAC aerosol dominated by BC has an AAE close to 1.0 while

the presence of BrC results in AAE larger than 1.0 (Andreae and Gelencsér, 2006). As described previously, AAE could be retrieved using a power law function as follows (Andreae and Gelencsér, 2006):

$$Abs(\lambda) = C \times \lambda^{-AAE} \tag{6}$$

where $C$ is a constant independent of wavelength.

Additionally, MAC could be used to reflect the light absorption capacity of aerosols. The MACs of BC and BrC at different wavelengths ($MAC_{BC}(\lambda)$ and $MAC_{BrC}(\lambda)$, respectively) were calculated with $Abs_{BC}(\lambda)$ and $Abs_{BrC}(\lambda)$ divided by the corresponding mass concentrations of BC and organic matter (OM), respectively:

$$MAC_{BC}(\lambda) = \frac{Abs_{BC}(\lambda)}{[BC]} \tag{7}$$

$$MAC_{BrC}(\lambda) = \frac{Abs_{BrC}(\lambda)}{[OM]} \tag{8}$$

where the mass concentration of OM was estimated by a factor of 1.8 times that of OC mass concentration (Turpin and Lim, 2001).

**2.5 Receptor model source apportionment**

The PMF version 5.0 (PMF5.0) from the US Environmental Protection Agency (Norris et al., 2014) was
applied to determine the contribution of various sources to aerosol light absorption. The principle of PMF has been described elsewhere (Paatero and Tapper, 1994). Briefly, PMF decomposes the initial dataset into a factor contribution matrix $G_{ik}$ ($i \times k$ dimensions) and a factor profile matrix $F_{kj}$ ($k \times j$ dimensions) and then iteratively minimizes the object function $Q$:

$$X_{ij} = \sum_{k=1}^{p} G_{ik}F_{kj} + E_{ij} \tag{9}$$

$$Q = \sum_{i=1}^{m} \sum_{j=1}^{n} \left(\frac{E_{ij}}{\sigma_{ij}}\right)^2 \tag{10}$$

where $X_{ij}$ was the value of the $j$th species in the $i$th sample; $E_{ij}$ described the model residual; and $\sigma_{ij}$ represented uncertainty, which was calculated as follows:

$$\sigma_{ij} = \begin{cases} \sqrt{(\text{error fraction} \times \text{concentration})^2 + (0.5 \times \text{MDL})^2}, & (\text{concentration} > \text{MDL}) \\ \frac{5}{6} \times \text{MDL}, & (\text{concentration} \leq \text{MDL}) \end{cases} \tag{11}$$

where MDL was the method detection limit and the error fraction was set to 10% (Rai et al., 2020). The uncertainties of the PMF5.0 results were evaluated by the following analyses: Bootstrap (BS), Displacement (DISP), and Bootstrap-displacement (BS-DISP). The BS analysis assesses the random errors in PMF solutions while DISP estimates rotational ambiguity. On the other hand, BS-DISP estimates both random errors and rotational ambiguity. A more detailed description of the three error estimation methods has been provided by Paatero et al. (2014) and Brown et al. (2015).

## 2.6 Analysis of air-mass trajectories

Cluster analysis of three-day backward air-mass trajectories was used to investigate the impact of transport pathways on Abs($\lambda$). The backward trajectories were calculated hourly with an arrival height of 500 m above ground level using the Hybrid Single-Particle Lagrangian Integrated Trajectory Model (Draxler and Rolph, 2003). The cluster analysis was performed according to the angle-based distance statistics method (Q. Wang et al., 2018a). Furthermore, a Concentration-weighted Trajectory (CWT) analysis based on the three-day backward trajectories was used to identify the potential source areas (Q. Wang et al., 2016). Finally, cluster and CWT analyses were performed using a GIS-based TrajStat software developed by Wang et al. (2009).

## 2.7 The Optical Properties of Aerosols and Clouds (OPAC) Model

The OPAC model was used to retrieve the following parameters: aerosol optical depth (AOD), single scattering albedo (SSA), and asymmetric parameter (AP). The parameters were important in estimating the radiative effect of aerosols. A detailed description of the OPAC software package was given by Hess et al. (1998). The measured mass concentrations of OC, EC, and water-soluble ions as well as the estimated mineral dust loading (=[Fe]/0.035) during the day were used in the OPAC model to estimate the optical parameters. Moreover, the BC number concentration in the OPAC model was constrained by the measured EC mass concentration. Although several water-soluble ions and mineral dust were obtained, they did not contain all the water-soluble and insoluble material. Therefore, based on the measured data,

the number concentrations of water-soluble and insoluble materials were tuned. This was done until the differences between the OPAC-derived light scattering, light absorption, and SSA versus the corresponding PAX-measured values were within 5% (Fig. S3). After the aerosol light extinction coefficient (sum of light scattering and absorption) was obtained, the AOD was estimated as follows (Hess et al., 1998):

$$\text{AOD} = \sum_j \int_{H_{j,min}}^{H_{j,max}} \sigma_{e,j}(h)dh = \sum_j \sigma_{e,j}^1 N_j(0) \int_{H_{j,min}}^{H_{j,max}} e^{-\frac{h}{Z_j}}dh \tag{12}$$

where $H_{j,max}$ and $H_{j,min}$ were the upper and lower boundaries in layer $j$; $\sigma_{e,j}$ was the surface aerosol light extinction coefficient in layer $j$; $h$ was the layer height; $\sigma_{e,j}^1$ represented the aerosol light extinction coefficient that was normalized to 1 particle cm$^{-3}$; $N_j$ was the number concentration in layer $j$; and $Z$ was the scale height. Furthermore, the OPAC-derived AODs were tuned to match the satellite-derived AODs (https://giovanni.gsfc.nasa.gov/giovanni, last access: January 2020) by altering the scale height in OPAC until the difference between them was within 5%. Owing to closure with AOD and anchoring of chemical composition, the assumptions in the OPAC model did not have a significant impact on the estimation of radiative effect in subsequent section 2.8 (Satheesh and Srinivasan 2006).

## 2.8 Estimations of radiative effect and heating rate

The LAC direct radiative effect (DRE) was estimated by the Santa Barbara DISORT (Discrete Ordinate Radiative Transfer) Atmospheric Radiative Transfer (SBDART) model in the shortwave spectral region of 0.25–4.0 μm. A detailed description of the SBDART model was given by Ricchiazzi et al. (1998). The AOD, SSA, and AP are essential input parameters in the SBDART model and were obtained from the OPAC model (see section 2.7). In addition to these, several other input parameters were included, namely the surface albedo, solar zenith angle, and profiles of atmospheric parameters. The surface albedo was derived from the Moderate Resolution Imaging Spectroradiometer (https://modis-atmos.gsfc.nasa.gov/ALBEDO/index.html, last access: January 2020). On the other hand, the solar zenith angle was calculated with a specific time and location (i.e., latitude and longitude) using a small code from the SBDART model. Furthermore, six standard atmospheric vertical profiles (i.e., tropical, mid-latitude summer, subarctic summer, mid-latitude winter, subarctic winter and US62) were embedded in

the SBDART model. They provided vertical distributions of temperature, pressure, water vapor, and ozone density (Ricchiazzi et al., 1998). In this study, the mid-latitude summer was selected to represent the situation of Sanya based on its classification as a mid-latitude region. Obregón et al. (2015) demonstrated that the SBDART model could provide a reliable estimation of radiative effect. Moreover, aerosol DRE was defined as the difference in the radiation flux (F) either at the Earth's surface or at the top of the atmosphere, respectively with and without the aerosol in the atmosphere:

$$DRE = (F \downarrow - F \uparrow)_{\text{with aerosol}} - (F \downarrow - F \uparrow)_{\text{without aerosol}} \tag{13}$$

where $\downarrow$ and $\uparrow$ represented the downward and upward flux, respectively. Atmospheric DRE was then estimated by the difference between the DRE at the top of the atmosphere and the Earth's surface. Further, the atmospheric heating rate ($\frac{\partial T}{\partial t}$, in unit of K d$^{-1}$) caused by LAC aerosols was estimated using the first law of thermodynamics and hydrostatic equilibrium as follows (Liou, 2002):

$$\frac{\partial T}{\partial t} = \frac{g}{C_p} \times \frac{\Delta F}{\Delta P} \tag{14}$$

where $\frac{g}{C_p}$ was the lapse rate and g stood for acceleration due to gravity while $C_p$ described the specific heat capacity of air at a constant pressure (1006 J kg$^{-1}$ K$^{-1}$). Additionally, $\Delta F$ was the atmospheric DRE contributed by LAC aerosols and $\Delta P$ represented the atmospheric pressure difference (300 hPa).

## 3 Results and discussion

### 3.1 Overview of Abs($\lambda$)

The AE33 absorption was first corrected using PAX measurement and a strong correlation (r = 0.96, $p <$ 0.01) between them was found (Fig. S4). A slope of 2.3 was regarded as the correction factor and was comparable to the values of 2.0–2.6 reported by previous studies using a similar method (Qin et al., 2018; Tasoglou et al., 2017; Wang et al., 2019b). This difference may mainly be related to the matrix scattering and lensing effects. The time series of corrected Abs($\lambda$) is shown in Fig. 1 and a statistical summary of the data presented in Table 1. The average Abs($\lambda$) were 15.7 ± 5.3, 11.4 ± 3.7, 9.7 ± 3.0, 8.3 ± 2.6, 7.0 ± 2.2, and 4.9 ± 1.5 Mm$^{-1}$ at 370, 470, 520, 590, 660, and 880 nm, respectively in the throughout the study.

However, it is noteworthy that such single-wavelength calibrations may overestimate Abs($\lambda$) at long wavelengths (i.e., $\lambda$ = 590, 660, and 880 nm) and underestimate it at short wavelengths (i.e., $\lambda$ = 370 and 470 nm) owing to the correction factor's dependence on wavelength (Kim et al., 2019). Compared to previous work, the Abs($\lambda$) in this study were lower than those obtained from urban areas in China and Europe (J. Wang et al., 2018; Liakakou et al., 2020). However, they were comparable to some rural and remote areas where anthropogenic activities were not intensive (Zanatta et al., 2016; Zhu et al., 2017). This suggests the possibility of a relatively small LAC burden in the atmosphere at Sanya during the study. Additionally, $Abs_{BC}(\lambda)$ contributed more than 77% to Abs($\lambda$) whereas the contribution of $Abs_{BrC}(\lambda)$ was less than 17% (Fig. 2). This was consistent with previous studies showing that BC was stronger at absorbing light compared to BrC at the near-ultraviolet to near-infrared wavelengths in the atmosphere (Massabò et al., 2015; Liakakou et al., 2020). However, laboratory studies reported that $Abs_{BrC}(\lambda)$ could exceed $Abs_{BC}(\lambda)$ at short wavelengths in fresh smoke from biomass burning, especially in the smoldering phase (Tian et al., 2019; Chow et al., 2018). Furthermore, the fraction of $Abs_{BC}(\lambda)$ increased with an increase in wavelength although the fraction of $Abs_{BrC}(\lambda)$ showed an inverse trend with a dramatic drop from 17% at 370 nm to 3% at 660 nm as shown in Fig. 2. This suggests a stronger light-absorbing capacity by BrC at short wavelengths compared to the long ones. With regard to the relationship between Abs($\lambda$) and carbonaceous composition, the $Abs_{BC}(\lambda)$ correlated well with EC mass concentration (r = 0.93, $p <$ 0.01, Fig. S5). However, a weak but significant correlation was observed between $Abs_{BrC}(\lambda)$ and OC mass concentration (r = 0.27–0.42, $p < 0.05$, Fig. S6). The results further confirmed that BC was the dominant light-absorbing material in LAC aerosols while OC contained more of non-light-absorbing carbon components compared to the light-absorbing ones.

**3.2 Source apportionment of Abs($\lambda$)**

To quantify the contributions of various sources to Abs($\lambda$), chemical species and $Abs_{pri}(\lambda)$ were simultaneously used as input parameters in the PMF5.0 model. Online $Abs_{pri}(\lambda)$ data was integrated to match each filter sampling time. The selected chemical species included carbonaceous matter (i.e., OC and EC), water-soluble cations (i.e., $Na^+$, $K^+$, and $Ca^{2+}$), elements (i.e., Ti, V, Mn, Fe, Ni, Cu, Zn, and Br), and hopanes. The mass concentrations of the chemicals are summarized in Table 2. Based on Eq. (5),

$Abs_{sec}(\lambda)$ accounted for less than 5% of $Abs(\lambda)$ (Table S1), suggesting a negligible impact of secondary formation on the light absorption capacity of aerosols during the study. Therefore, the uncertainty caused by using only $Abs_{pri}(\lambda)$ in the model could be put to rest in the absence of an effective way to identify the source of secondary BrC.

Moreover, two to seven factors were selected to initiate the PMF5.0 run. Due to the additional factors, the $Q/Q_{exp}$ ratio decreased with the increased number of factors as shown in Fig. S7. The decrease in $Q/Q_{exp}$ was large when the factor number changed from 2 to 3 and 3 to 4 but stabilized as the factor number grew larger than 4, indicating that four factors may be the optimal solution. After multiple runs of the PMF5.0 model, four factor sources including ship emissions, motor vehicle emissions, biomass

burning, and fugitive dust were finally identified (Fig. 3a). Additionally, the modeled $Abs_{pri}(\lambda)$ at different wavelengths showed strong correlations with the measured $Abs(\lambda)$ (r = 0.82–0.89, $p < 0.01$, Fig. S8). The slopes of 0.92–0.98 were consistent with the absorption fractions of $Abs_{pri}(\lambda)$ estimated by the BC-tracer method combined with the MRS approach (Table S1). The scaled residuals for each species varied between -3 and +3.

The uncertainty of each factor profile was further evaluated using BS, DISP, and BS-DISP. The BS results showed that the reproducibility of each source factor was larger than 80% (Table S2), indicating good stability. Therefore, this suggested that the four source factors were appropriate. No swaps occurred in DISP, indicating the stability of the selected solution. Furthermore, all BS-DISP runs were successful. Overall, these results pointed to the efficiency of the PMF5.0 model in performing optical source

apportionment.

The first source factor was characterized by large proportions of V, Ni, and hopanes as well as moderate amounts of OC, EC, $Na^+$, $K^+$, Cu, and $Abs_{pri}(\lambda)$ as shown in Fig. 3a. V and Ni were associated with oil fuel combustion (Moreno et al., 2010) and their ratio (V/Ni) can be used to further identify ship engine emissions, which has a typical range of 2.5–4.0 (Cesari et al., 2014). The estimated V/Ni ratio was 3.4 in

this source factor, consistent with the previously established range of ship engine emissions. Since hydrocarbons are the major components of ship engine oil, hopanes, OC, and EC can be produced as byproducts of the combustion process. Therefore, this source factor was assigned to ship emissions. The second source factor was associated with large amounts of Cu, Zn, and Br as well as moderate proportions

of hopanes, EC, Ti, and $Abs_{pri}(\lambda)$. Previous studies confirmed that hopanes, Br, and EC were typically present in vehicle exhaust particles (Huang et al., 1994; Sheesley et al., 2009). Additionally, Zn and Cu were associated with lubricant and metal brake wear (Lin et al., 2015). Therefore, this source factor was allocated to motor vehicle emissions. The third source factor was dominated by high proportions of $K^+$, OC, EC, and $Abs_{pri}(\lambda)$ which was an obvious feature of biomass burning (Forello et al., 2019). Finally, the fourth source factor was characterized by large amounts of several crustal materials such as $Ca^{2+}$, Ti, Fe, and Mn and was identified as fugitive dust.

Notably, biomass burning occupied the largest proportion of light absorption ($Abs_{biomass}(\lambda)$) at 32–44% of $Abs_{pri}(\lambda)$ as shown in Fig. 3a. Sanya is coastal city with heavy maritime traffic (e.g., the cargo handling capacity was larger than 5.8 million tons in 2017 at Sanya port, http://tjj.sanya.gov.cn/tjjsite/2019nnj/tjnj.shtml, in Chinese) therefore absorption of ship emissions ($Abs_{ship}(\lambda)$) also had a significant contribution to $Abs_{pri}(\lambda)$ (30–39%). The contribution of motor vehicle emissions ($Abs_{vehicle}(\lambda)$ = 17–24% of $Abs_{pri}(\lambda)$)) was much lower than that of biomass burning and ship emissions. Moreover, the absorption of fugitive dust ($Abs_{dust}(\lambda)$) occupied less than 10% of $Abs_{pri}(\lambda)$, consistent with previous reports where it was identified as a minor contributor in the atmosphere (Yang et al., 2009; Zhao et al., 2019). This small absorptive fraction may be attributed to the low proportion of light-absorbing iron oxides in the atmosphere. Furthermore, the $Abs_{pri}(\lambda)$ of different sources all decreased with increased wavelength (Fig. 3b) although their relative contributions displayed distinct trends (Fig. 3a). The fraction of $Abs_{ship}(\lambda)$ and $Abs_{vehicle}(\lambda)$ increased with an increase in wavelength while a reverse trend was observed in the $Abs_{biomass}(\lambda)$ fraction. This discrepancy can be explained by the large amount of BrC present in biomass-burning emissions which can result in more light absorption at short wavelengths relative to the long ones.

To identify the possible source areas that affected $Abs_{pri}(\lambda)$, CWT analysis was performed based on the three-day backward trajectories. Large CWT values were mainly concentrated in the South China Sea (Fig. S9), highlighting the effect of ship emissions on aerosol light absorption. Additionally, the three-day backward trajectories were grouped into four cluster-mean trajectories to investigate the impact of different sources on $Abs_{pri}(\lambda)$ (Fig. 4). The air masses associated with Cluster #1 originated from the South China Sea. The $Abs_{ship}(\lambda)$ was the largest contributor in this cluster constituting 44–45% of $Abs_{pri}(\lambda)$

due to the high vessel traffic density over the South China Sea, consistent with the CWT results. Cluster #1 accounted for about 44% of the total trajectories, suggesting that Sanya was subjected to the influence of ship exhaust-related LAC aerosols transported from the South China Sea. It is noteworthy that the diurnal pattern of $Abs_{pri}(\lambda)$ showed typically high values in the mornings and evenings (Fig. S10). This was attributed to the daily anthropogenic activities and variations in height of the planetary boundary layer. Given that the air masses from the South China Sea are unable to carry pollutants from biomass burning, motor vehicles, and fugitive dust, these sources were possibly mainly influenced by local emissions.

Cluster #2 originated from the South China Sea near the Indochina Peninsula and accounted for 35% of the total trajectories. The $Abs_{ship}(\lambda)$ was also vital in this cluster, accounting for 34–37% of $Abs_{pri}(\lambda)$. Fig. S10 shows that the $Abs_{pri}(\lambda)$ of Cluster #2 displayed a similar diurnal trend as that of Cluster #1. Considering that the air masses of Cluster #2 also originated from the South China Sea, the sources except for ship emissions were mainly influenced by local discharge. A small number of air masses were grouped into Cluster #3 and Cluster #4, accounting for only 6% and 15% of total trajectories, respectively. Cluster #3 originated from southern Burma and passed over Thailand, Laos, and Vietnam. On the other hand, Cluster #4 had the longest cluster-mean trajectory which originated and passed through the coastal areas of South-eastern China. Biomass burning was the dominant contributor to $Abs_{pri}(\lambda)$ in both clusters, with 62–69% for Cluster #3 and 56–64% for Cluster #4. Moreover, the $Abs_{biomass}(\lambda)$ of Cluster #3 and Cluster #4 were 1.8–4.4 times higher than those of Cluster #1 and Cluster #2. Since the $Abs_{biomass}(\lambda)$ from Cluster #1 and Cluster #2 were mainly attributed to local emissions, the higher values in Cluster #3 and Cluster #4 may have been influenced by the long-range transport of biomass burning from Southeast Asia and South-eastern China, where there was a large number of fire incidences (Fig. 4).

### 3.3 Source-dependent optical properties of LAC aerosols

According to a power law function (Fig. 5), the average LAC AAE (1.41, Table 1) was greater than unity during the study, indicating the presence of both BC and BrC in the atmosphere. In addition, the estimated AAE of motor vehicle emissions ($AAE_{vehicle}$) was 0.96. This was close to the previously reported range of 0.9–1.1 obtained from ambient observations using the radiocarbon method or vehicle exhaust-related

source experiments (Sandradewi et al., 2008; Zotter et al., 2017; Chow et al., 2018). This narrow range of $AAE_{vehicle}$ obtained by various studies suggests that the spectral dependence of vehicle exhaust-related LAC absorption was less affected by atmospheric processes. Furthermore, the AAE of ship emissions ($AAE_{ship} = 1.06$) was similar to that of $AAE_{vehicle}$. These low spectral dependences of light absorption

indicate that BC was the dominant compound in LAC aerosols from ship and motor vehicle emissions. Compared to marine engine emissions, the $AAE_{ship}$ obtained in this study was consistent with the values derived from marine gas oil and diesel fuel emissions ($1.0 \pm 0.1$) but was lower than heavy fuel oil exhaust ($1.7 \pm 0.2$) (Corbin et al., 2018). This indicates that Sanya may be influenced more by ships using distillate rather than heavy fuel oil.

The AAE of biomass burning ($AAE_{biomass} = 1.75$) was larger than that from ship and motor vehicle emissions. This implied the presence of BrC in LAC aerosols derived from biomass-burning in addition to BC. The observation corroborated with previous studies which showed that BrC was mainly derived from biomass burning rather than fossil fuels in the atmosphere (Laskin et al., 2015). Additionally, chamber studies showed that the AAEs of fresh smoke from biomass burning varied largely (e.g., 1.64–

3.25) depending on the type of biomass and their burning efficiencies (Tian et al., 2019). The $AAE_{biomass}$ from this study was close to those (1.7–1.9) from the atmosphere constrained by the radiocarbon method (Sandradewi et al., 2008; Zotter et al., 2017). Given that the approach used in this study could retrieve the source-specific AAEs in the atmosphere, it can also improve the performance of those optical source apportionment models based solely on optical data.

Owing to the dominance of BC in ship and motor vehicle emissions, only $MAC_{BC}(\lambda)$ was estimated for these two sources. The results of optical source apportionment revealed that the estimated $MAC_{BC}(\lambda)$ of motor vehicle emissions ($MAC_{BC,vehicle}(\lambda)$) were close to the values of uncoated BC particles at different wavelengths (Fig. 6). This indicated that vehicle exhaust-related BC particles were mainly associated with local emissions and underwent minor atmospheric aging processes. In contrast, the $MAC_{BC}(\lambda)$ of ship

emissions ($MAC_{BC,ship}(\lambda)$) was 1.4–1.6 times larger than that of the uncoated ones (Fig. 6). This implied that ship exhaust-related BC particles were prone to substantial aging during transit from the ocean. Freshly emitted BC particles from fossil fuels tend to mix externally with other substances and become internally-mixed ones after aging (Xing et al., 2020). It was therefore unexpected for the obtained

$MAC_{BC,ship}(\lambda)$ to have a similar value as that of marine engine emissions reported by Corbin et al. (2018) (7.8 m$^2$ g$^{-1}$ at 780 nm, extrapolated to the same wavelengths in this study by assuming an $AAE_{BC} = 1.1$). Consequently, more work is needed to understand the large $MAC_{BC}(\lambda)$ values from marine engine emissions.

Since LAC aerosols derived from biomass burning comprised of both BC and BrC, the $MAC_{BC}(\lambda)$ and $MAC_{BrC}(\lambda)$ were retrieved based on the results of optical source apportionment. The findings revealed that the $MAC_{BC}(\lambda)$ of biomass burning ($MAC_{BC,biomass}(\lambda)$) was larger than $MAC_{BC,vehicle}(\lambda)$ as shown in Fig. 6. This was consistent with previous studies showing a stronger capacity to absorb light by BC from biomass burning compared to that from motor vehicle emissions (Qiu et al., 2014; Q. Wang et al., 2018b).

Moreover, the $MAC_{BC,biomass}(\lambda)$ was smaller than $MAC_{BC,ship}(\lambda)$, suggesting a stronger ability to absorb light by BC particles from ship exhaust. A broader implication of this observation is that more focus should be put on BC particles related to ship emissions due to their impact on climate given the increase in shipping activities globally.

The $MAC_{BrC}(\lambda)$ of biomass burning ($MAC_{BrC,biomass}(\lambda)$) was highly dependent on wavelength, with 0.9
m$^2$ g$^{-1}$ at $\lambda = 370$ nm but dropped close to zero (0.02 m$^2$ g$^{-1}$) at $\lambda = 660$ nm (Fig. 6). Additionally, the $MAC_{BrC,biomass}(\lambda)$ was several times to two orders of magnitude lower than $MAC_{BC}(\lambda)$ from different sources, suggesting that BC had a stronger ability to absorb light compared to BrC. Notably, the $MAC_{BrC}(\lambda)$ obtained in this study lied within the range reported by previous investigations although with differences among studies (Wang et al., 2019b; Cho et al., 2019). The differences in $MAC_{BrC}(\lambda)$ may
partly be related to biomass types and their burning efficiencies as well as the aging processes of BrC in the atmosphere. In addition, the use of different BrC substitutes (e.g., OM, organic aerosol, or water-soluble organic carbon) may have impacted the calculation of $MAC_{BrC}(\lambda)$. Compared to previous laboratory studies, the $MAC_{BrC,biomass}(\lambda)$ obtained here was smaller than that of fresh smoke from biomass burning (Zhong and Jang, 2014; Pandey et al., 2016). Given that photobleaching is an effective way of
turning BrC into a transparent organic substance (Laskin et al., 2015), the smaller atmospheric $MAC_{BrC,biomass}(\lambda)$ observed in this study may be attributed to the elimination of organic chromophores induced by the bright sunlight at Sanya.

The MAC links the mass of a LAC aerosol to its ability to absorb light and is an important parameter in climate models to evaluate global or regional radiative effects of LAC aerosols. An equal MAC from different sources is often assumed in climate models (Bond et al., 2013) because identification of source-specific MACs in the atmosphere is still a challenge. However, this assumption can lead to uncertainties due to distinct MACs from various sources (e.g., $MAC_{BC,ship}(\lambda) > MAC_{BC,biomass}(\lambda) > MAC_{BC,vehicle}(\lambda)$ in this study). The chemical composition-based optical source apportionment approach may provide a potential solution to this challenge. Nonetheless, more source-specific MACs in different areas and seasons are needed in future studies to gauge the accuracy of climate models. Moreover, this approach may minimize the uncertainties of BC source apportionment using the 'aethalometer model' (Sandradewi et al., 2008; Zotter et al., 2017) due to the assumption of equal AAE and MAC from different sources.

## 3.4 Impacts of LAC aerosols on radiative effect

Fig. 7 shows the source-specific LAC DRE during the study. The LAC DRE varied from -5.5 to -1.6 W m$^{-2}$ at the Earth's surface with an average cooling effect of -3.2 ± 1.0 W m$^{-2}$. In contrast, the LAC aerosols produced a warm effect of +1.5 ± 0.5 W m$^{-2}$ at the top of the atmosphere with a range of +0.8 to +2.8 W m$^{-2}$ suggesting a net energy gain. The presence of LAC aerosols enhanced aerosol DRE at the top of the atmosphere by 62% compared to the results of light scattering by aerosols only. Moreover, the difference between LAC DRE at the top of the atmosphere and the Earth's surface gave the atmospheric DRE (a net atmospheric absorption) of +4.7 ± 1.5 W m$^{-2}$ and could generate a heating rate of 0.13 ± 0.04 K day$^{-1}$. With regard to LAC absorption sources, biomass burning was the largest contributor to LAC DRE at -1.5 ± 0.5 W m$^{-2}$ and +0.7 ± 0.2 W m$^{-2}$ on the surface of the Earth and the top of the atmosphere, respectively. In addition, BrC from biomass burning had a less contribution to LAC DRE compared to BC from the same source. However, the presence of BrC reinforced the LAC DRE of biomass burning by 21% as opposed to BC only, suggesting a substantial radiative effect from BrC aerosol. Additionally, the LAC DRE were -1.1 ± 0.4 W m$^{-2}$ and +0.5 ± 0.2 W m$^{-2}$ for ship emissions and -0.6 ± 0.2 W m$^{-2}$ and +0.3 ± 0.1 W m$^{-2}$ for motor vehicle emissions on the Earth's surface and the top of the atmosphere, respectively. The LAC DRE contributed by ship and motor vehicle emissions was mainly caused by BC aerosol. Although a larger BC atmospheric DRE was observed for biomass burning, ship emissions showed an equivalent

capacity of radiative effect (0.5 (W m$^{-2}$) ($\mu$g m$^{-3}$)$^{-1}$) from per unit BC mass concentration generating atmospheric DRE. In contrast, motor vehicle emissions had a smaller value of 0.3 (W m$^{-2}$) ($\mu$g m$^{-3}$)$^{-1}$. Furthermore, the atmospheric heating rate of LAC aerosols was similar for biomass burning (0.06 $\pm$ 0.02 K day$^{-1}$) and ship emissions (0.05 $\pm$ 0.01 K day$^{-1}$) but larger than that produced by motor vehicle emissions (0.03 $\pm$ 0.01 K day$^{-1}$). This further highlighted the importance of LAC aerosols from ship exhaust in atmospheric heating.

## 4 Conclusions

In this study, the optical properties and radiative effect of LAC aerosols in Sanya, a Chinese tropical marine monsoon climate zone, were explored. The study found that light absorption caused by primary emissions was the main contributor to LAC absorption while secondary processes played a minor role. Moreover, BC aerosol (> 77%) contributed more to Abs$_{LAC}$($\lambda$) compared to BrC (< 17%). Through a combination of chemical species and multi-wavelength absorption in a positive matrix factorization model, it was shown that biomass burning had the highest contribution to Abs$_{pri}$($\lambda$) (32–44%) followed by ship (30–39%) and motor vehicle emissions (17–24%). Fugitive dust had the lowest contribution (< 10%). Furthermore, cluster analysis of three-day backward trajectories showed that ship emissions were the major contributors to Abs$_{pri}$($\lambda$) when the air-masses originated from the South China Sea whereas biomass burning dominated in other directions.

Moreover, source-specific AAE showed a similarity between ship and motor vehicle emissions (1.06 versus 0.96). The low spectral dependence of light absorption indicated that LAC aerosols were dominated by BC in ship and motor vehicle emissions. In contrast, a large AAE of 1.75 was found in biomass burning, indicating the presence of both BC and BrC. Additionally, source-specific MAC showed that BC particles from ship emissions had the strongest light-absorbing capacity followed by biomass burning and motor vehicle emissions. Compared to BC MAC, the BrC MAC of biomass burning was smaller with a value of 0.9 m$^2$ g$^{-1}$ at $\lambda$ = 370 nm but dropped to 0.02 m$^2$ g$^{-1}$ at $\lambda$ = 660 nm. The radiative transfer model also showed that the atmospheric DRE caused by LAC aerosols was +4.7 $\pm$ 1.5 W m$^{-2}$ during the study and corresponded to a heating rate of 0.13 $\pm$ 0.04 K day$^{-1}$. The presence of BrC reinforced the LAC DRE of biomass burning by 21% as compared to BC only. Finally, ship emissions showed an

equivalent capacity to produce radiative effect (0.5 (W m$^{-2}$) ($\mu$g m$^{-3}$)$^{-1}$) from per unit BC mass concentration generating atmospheric DRE. In contrast, motor vehicle emissions had a smaller value of 0.3 (W m$^{-2}$) ($\mu$g m$^{-3}$)$^{-1}$.

*Data availability*. All data described in this study are available upon request from the corresponding authors.

*Supplement*. The supplement related to this article is available online.

*Author contributions*. QW, JC, and YH designed the campaign. PW and YZ provided the observation site and assisted with field sampling and measurements. WD and TZ conducted the chemical analyses. HL ran the PMF5.0 and SBDART model. JT performed the cluster analysis of air-mass trajectories. WZ provided the ArcGIS maps. QW conducted the data analysis and wrote the article with input from all co-authors.

*Competing interests*. The authors declare that they have no conflict of interest.

*Acknowledgments*. This work was supported by the Strategic Priority Research Program of Chinese Academy of Sciences (XDB40000000), the Youth Innovation Promotion Association of the Chinese Academy of Sciences (2019402), the Hainan Natural Science Foundation High-level Talent Project (2019RC243), and the Science and Technology Cooperation Project of Sanya (2018YD14 and 2012YD38).

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

**Table 1.** Summary of light absorption at different wavelengths (Abs($\lambda$), $\lambda$ = 370, 470, 520, 590, 660, and 880 nm) and Absorption Ångström Exponent (AAE) of different emission sources.

| Parameter[a] | Average | Standard deviation |
|---|---|---|
| Abs(370) (Mm$^{-1}$) | 15.7 | 5.3 |
| Abs(470) (Mm$^{-1}$) | 11.4 | 3.7 |
| Abs(520) (Mm$^{-1}$) | 9.7 | 3.0 |
| Abs(590) (Mm$^{-1}$) | 8.3 | 2.6 |
| Abs(660) (Mm$^{-1}$) | 7.0 | 2.2 |
| Abs(880) (Mm$^{-1}$) | 4.9 | 1.5 |
| $AAE_{total}$ | 1.41 | 0.05 |
| $AAE_{ship}$ | 1.06 | 0.03 |
| $AAE_{biomass}$ | 1.75 | 0.06 |
| $AAE_{vehicle}$ | 0.96 | 0.06 |

[a]$AAE_{total}$ represents the AAE caused by total light-absorbing aerosols while $AAE_{ship}$, $AAE_{biomass}$, and $AAE_{vehicle}$ are AAE from ship emissions, biomass burning, and motor vehicle emissions, respectively.

**Table 2.** The average mass concentrations of $PM_{2.5}$, carbonaceous matter, water-soluble ions, inorganic elements, and organics during the campaign.

| Types | Species | Average | Standard deviation |
|---|---|---|---|
| $PM_{2.5}$ ($\mu g\ m^{-3}$) | | 14.3 | 4.2 |
| Carbonaceous matter ($\mu g\ m^{-3}$) | organic carbon | 2.7 | 1.1 |
| | elemental carbon | 0.8 | 0.3 |
| Water-soluble ions ($\mu g\ m^{-3}$) | $Na^+$ | 0.5 | 0.2 |
| | $NH_4^+$ | 0.6 | 0.4 |
| | $K^+$ | 0.2 | 0.1 |
| | $Mg^{2+}$ | 0.05 | 0.02 |
| | $Ca^{2+}$ | 0.2 | 0.1 |
| | $Cl^-$ | 0.23 | 0.2 |
| | $NO_3^-$ | 0.6 | 0.3 |
| | $SO_4^{2-}$ | 3.5 | 1.2 |
| Inorganic elements ($ng\ m^{-3}$) | Ti | 13.1 | 9.7 |
| | V | 2.4 | 1.4 |
| | Mn | 5.1 | 2.7 |
| | Fe | 127.3 | 78.9 |
| | Ni | 1.1 | 0.6 |
| | Cu | 28.0 | 14.4 |
| | Zn | 16.6 | 11.1 |
| | Br | 2.6 | 2.0 |
| Organics ($ng\ m^{-3}$) | hopanes | 0.2 | 0.05 |

**Figure Captions**

**Figure 1.** Time series of hourly averaged light absorption at different wavelengths (Abs($\lambda$), $\lambda$ = 370, 470, 520, 590, 660, and 880 nm). The different types of horizontal lines represent the four clusters of air masses.

**Figure 2.** Light absorption fractions of BC, BrC, and MD in the total light-absorbing aerosols. BC = black carbon; BrC = brown carbon; MD = mineral dust.

**Figure 3.** (a) Contributions of the four sources to each species from the positive matrix factorization model and (b) the light absorption of primary aerosols from each source at different wavelengths (Abs$_{pri}$($\lambda$), $\lambda$ = 370, 470, 520, 590, 660, and 880 nm) during the study.

**Figure 4.** Contribution of different sources to light absorption of primary aerosols in each three-day backward-trajectory cluster during the campaign at Sanya. The map was drawn using ArcGIS software. The base map is the World Topographic Map from © ESRI (Environmental Systems Research Institute, Inc.) (www.arcgis.com/home/item.html?id=30e5fe3149c34df1ba922e6f5bbf808f).

**Figure 5.** The light absorption (Abs($\lambda$)) of light-absorbing carbonaceous (LAC) aerosols from ship emissions, traffic emissions, and biomass burning. The dash line is power law fit.

**Figure 6.** The source-specific mass absorption cross section (MAC) of black carbon (BC) and brown carbon (BrC) at different wavelengths. The MAC of uncoated BC particles at each wavelength are extrapolated from 7.5 m$^2$ g$^{-1}$ at 550 nm (Bond and Bergstrom, 2006) by assuming an BC absorption Ångström exponent of 1.1.

**Figure 7.** Direct radiative effect (DRE) of light-absorbing carbonaceous (LAC) aerosols from biomass burning, ship emissions, and motor vehicle emissions. The error bar represents one standard deviation. ES, TOA, ATM represent the DRE at the Earth's surface, the top of the atmosphere, and in the atmosphere, respectively.

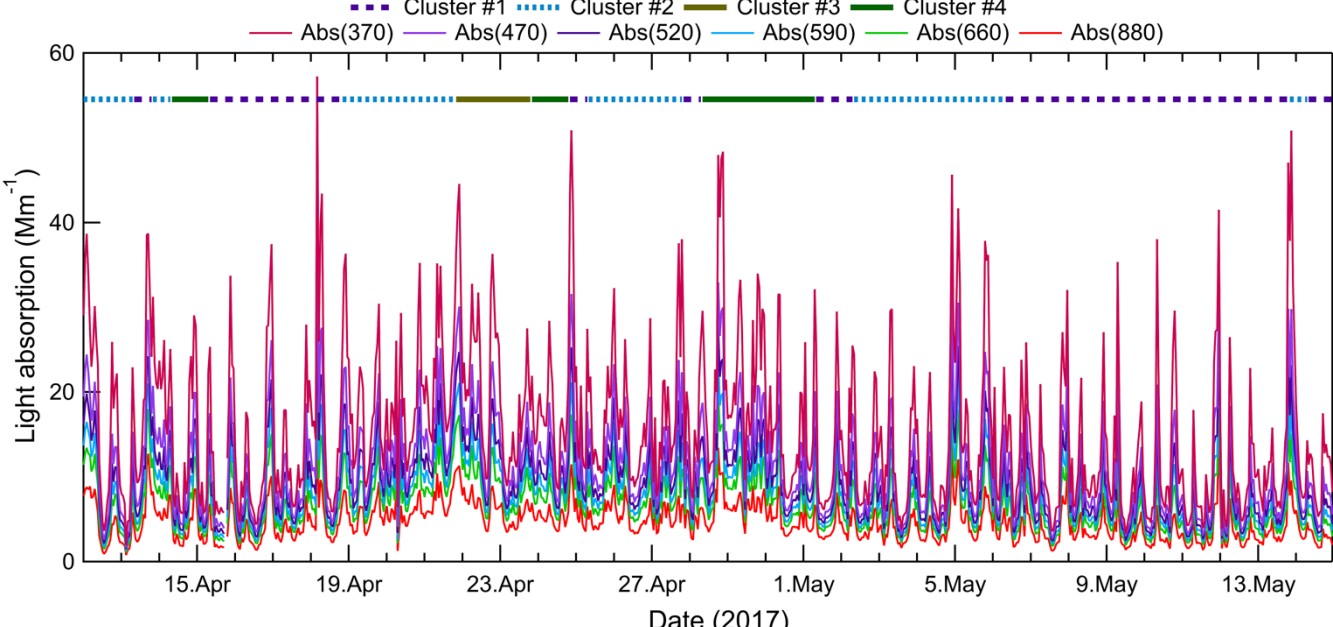

**Figure 1.**

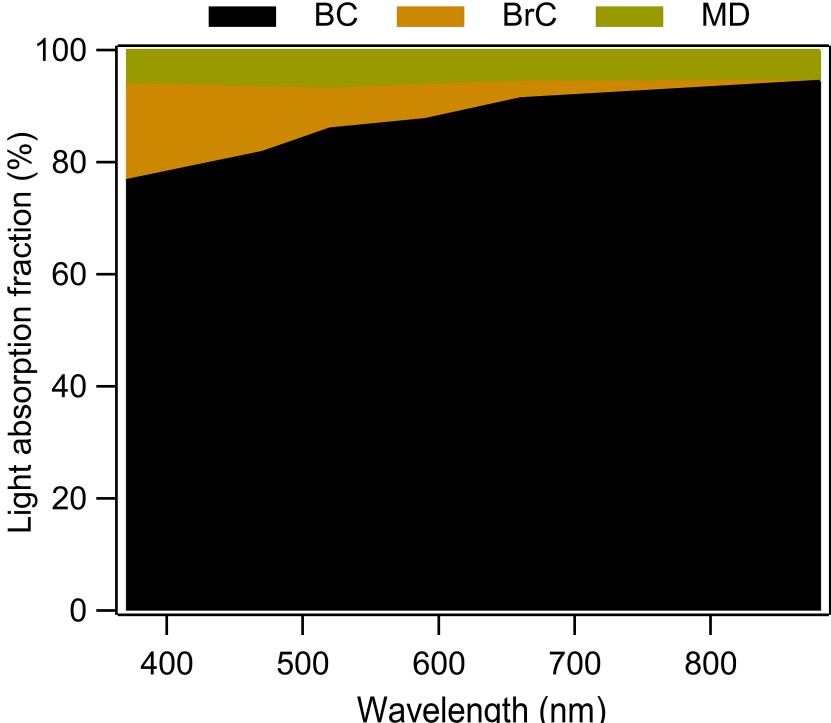

**Figure 2.**

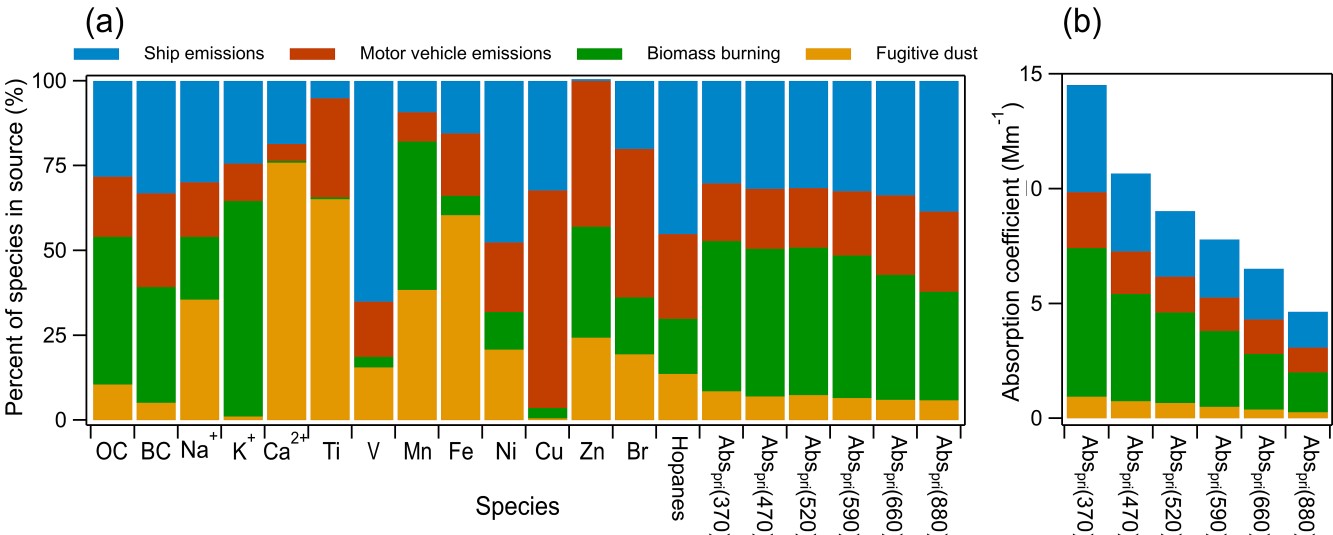

**Figure 3.**

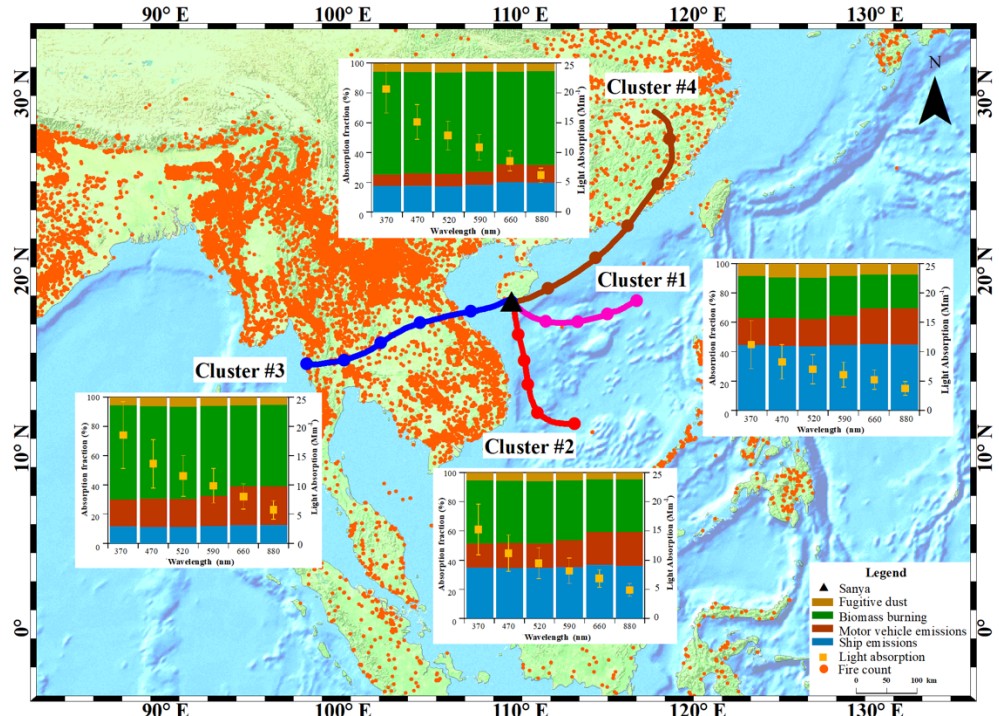

**Figure 4.**

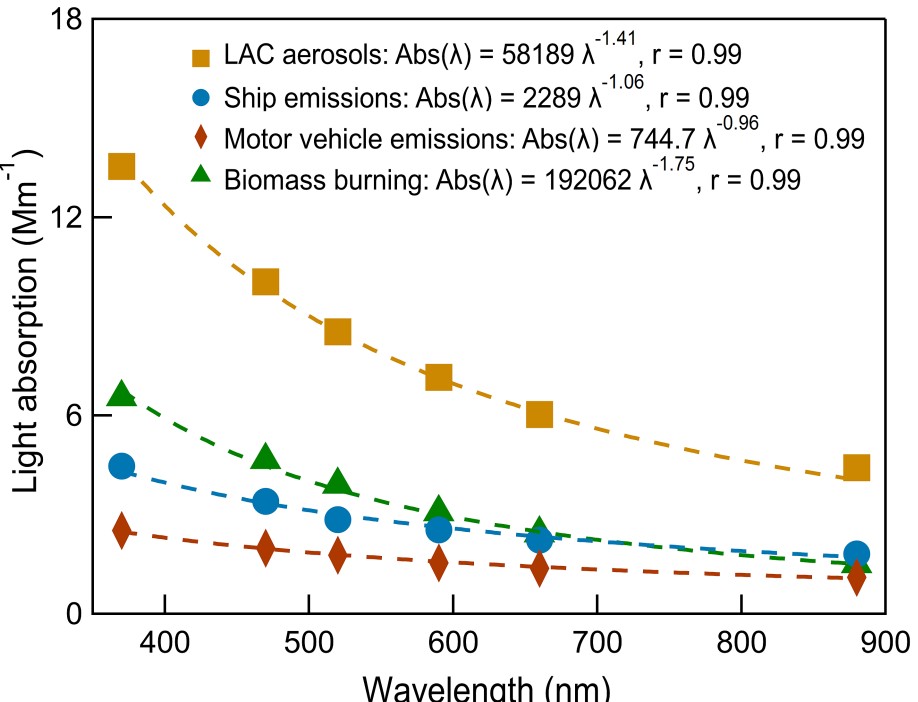

**Figure 5.**

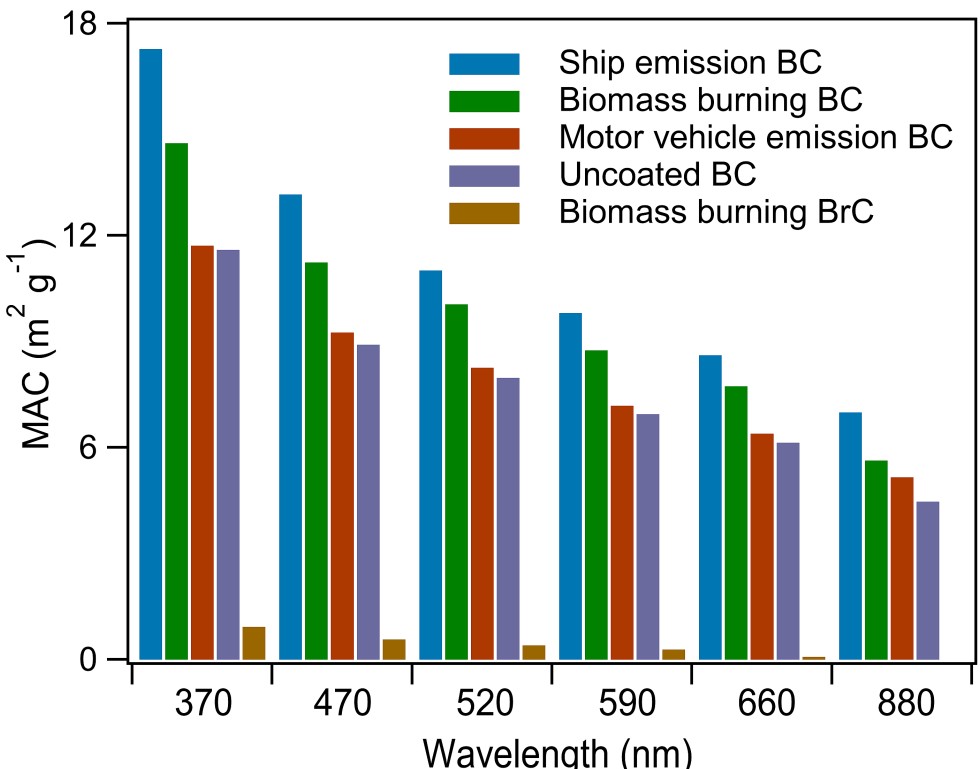

**Figure 6.**

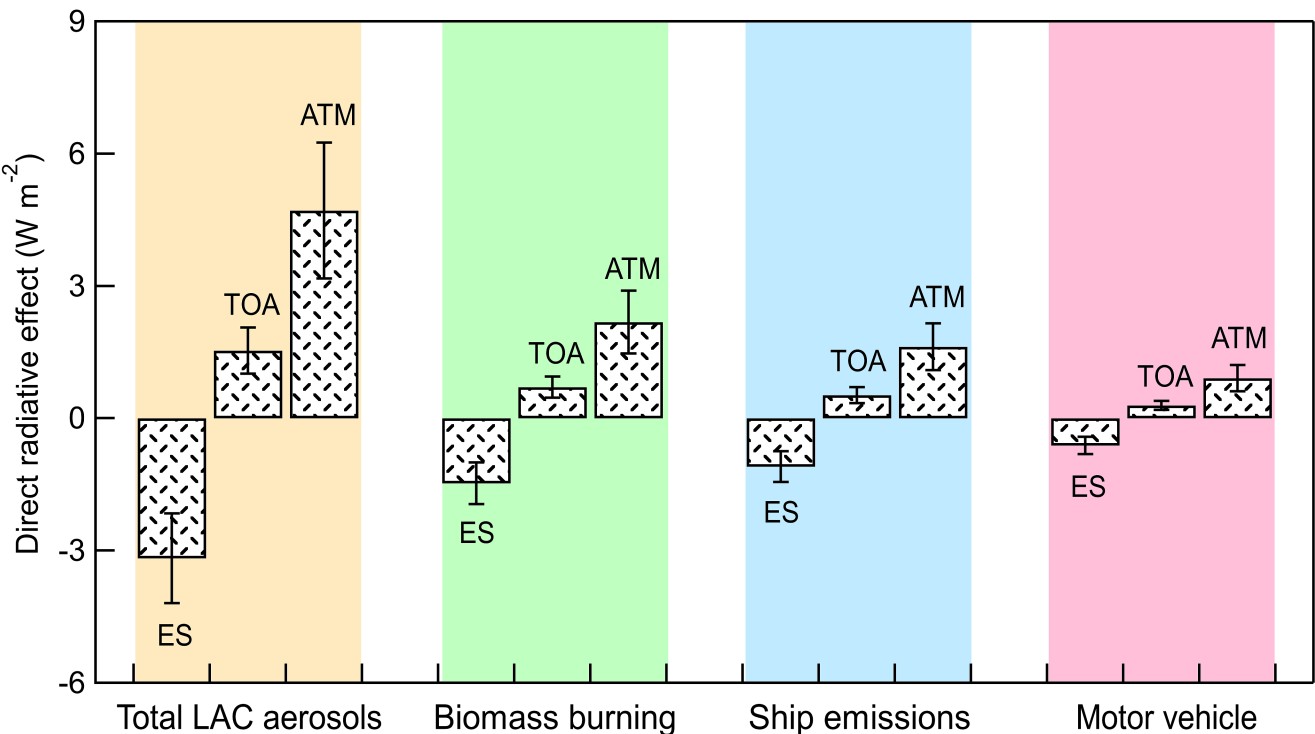

**Figure 7.**