# Peer review of "Optical source apportionment and radiative effect of light-absorbing carbonaceous aerosols in a tropical marine monsoon climate zone: The importance of ship emissions"

_Atmospheric Chemistry and Physics, 2020_

## Referee Comment (RC1) · Anonymous Referee #1 · 5 Jul 2020

General comments

The authors present an experimental study of aerosols collected from Hainan Island, South China. The analysis includes absorption coefficients, mass concentrations of black carbon, organic carbon, inorganic elements, and water-soluble cations and anions. Major findings include the source apportionment of the total absorption coefficient and contribution to radiative forcing. The study shows the importance of considering ship emissions in forcing calculation. Overall, the manuscript presents interesting data and analysis shows merit.

[Figure]

Specific comments

1. This study uses AE-33 and PAX to measure absorption. AE-33 provides the mass of absorbing aerosols as final products. Previous studies have reported the calculation of absorption coefficients from AE-33 mass concentration. However, for the sake of completeness, I would recommend to include those steps in supplementary.

2. This study used a Nafion dryer to reduce the RH of particles collected. These dryers are known to minimize particle concentration during the drying process. Is there any data on the % of particle loss within the dryer?

3. The Nafion dryers were connected to Aethalometers only? Aethalometer data is less susceptible to RH. But the PAX data can be influenced by high RH. Was there a dryer connected to PAX?

4. There was a PM2.5 cyclone for Aethalometer and no cyclone for PAX. I remember the penetration efficiency of PAX reduces drastically after 1 micrometer. So, both instruments were measuring different size-cutoff particles.

5. What is the area of quartz filters used?

6. What is the flow rate of the high-volume sampler?

7. One major shortcoming in this study is the absence of 'lensing effect' while calculating absorption. Studies have shown that the lensing effect can contribute to significant absorption. Since Aethalometer uses a filter tape to collect particles, one can assume the core-shell structure of particles (the reason for lensing effect) gets destroyed. But the absorption from PAX will have contributions from the lensing effect. The slope pf 2.29 in Figure S3 might include the lensing effect. Since the experimental setup used in this study does not measure the absorption of core-shell and core separately, it will be difficult to distinguish the contribution from the lensing effect. I would suggest the authors include this possibility in text.

8. Figure 1a – shows the apportionment of Abs, and the same is repeated as Figure

1b. Removing the repeated portion from 1a would give better visibility to it.

9. Page 2, line 13- Optical properties of LAC is not just related to its source. It also depends on the atmospheric conditions and secondary processing.

10. Page 4, line 3 – Educational and residential areas will have their pollution sources such as vehicles, cooking, etc.

11. Page 5, Paragraph 1 – The whole paragraph is about the analysis of filters collected. It must be specified initially.

12. Page 7, line 4 – Which PMF system was used for the analysis? I guess US EPA PMF 5.0! It needs to be mentioned with a reference.

13. Page 9, line 24 – Error bars on Y-axis needed. Since the X-axis is from filters (12-hour sample) and the Y-axis is the average of the same from AE-33 Abs, the error bars are required to see the spread of data.

14. Page 12, line 1 – The cluster 2 back trajectory doesn't touch the Vietnam cost to influence the biomass burning. Was there a spread towards land for this cluster?

Technical corrections

15. Page 2, line 26 – Don't use 'firstly'. 'First' is fine.

16. Page 4, line 10 – 'As described previously' – It is not described anywhere before.

---

## Referee Comment (RC2) · Anonymous Referee #2 · 6 Jul 2020

This study performed ground measurements in a small city at the very south end of continental China along the South China Sea. The authors attempted to attribute the sources from the composition and absorption measurements using receptor models, however the analysis of the entire study is rough and the conclusions are vague for the current version.

-Many pieces of essential work are missing, which should all appear in the main figures. I only list a few examples: the time series as classified by clusters, the diurnal variations of absorption for each cluster (to exclude the possible local sources), complete

statistics of all parameters are required (BC mass, AAE, PM, compositions). Please do a complete and sound analysis and just show it. Otherwise, the conclusions are based on nowhere.

-many issues here regarding the source attribution. The local source influence needs to be clearly excluded, or by some way to show it is of minor influence compared to the regional sources you stated. Not clear with the definition of secondary substance, most organic and mineral should also be primary? Where did the dust come from, I don't think there was any dust sources rather than some sea salt. The shipping emissions are not really supported with any other external data source (I don't know what is that because I can't see anything from the current analysis done so far).

-The PMF analysis seems not quite convincing, could you provide more details about the scenarios. And even so, would you really believe to incorporate the spectral absorption in parallel with the offline composition will really give some physical meaning? There seems no signature of sources on the absorption. The time resolution is different between online and offline measurements, did you just average the online data into a very low time interval.

-MAC of organic should be normalized by organic matter (including all elements) not only organic carbon. The MAC of organic here doesn't mean anything.

- I am not convinced with the forcing calculation and the directly correlated heating. I don't think you really need to make that calculation as the main job of this study is to get the absorption attribution properly. The forcing largely replies on the vertical distribution of AOD and SSA, which you don't really have such information, which is beyond the scope of this study though.

-No need such redundant description for the trajectory clustering, as this is not your original work and has been so widely used previously, which was just output using some user-friendly software.

СЗ

---

## Author Comment (AC1) · 10 Sep 2020

General comments

The authors present an experimental study of aerosols collected from Hainan Island, South China. The analysis includes absorption coefficients, mass concentrations of black carbon, organic carbon, inorganic elements, and water-soluble cations and anions. Major findings include the source apportionment of the total absorption coefficient and contribution to radiative forcing. The study shows the importance of considering ship emissions in forcing calculation. Overall, the manuscript presents interesting data and analysis shows merit.

**Response:** We thank the reviewer for his/her valuable suggestions, and it is useful for improving our manuscript. We have made modifications accordingly based on the reviewer's comments. Below are point-to-point responses.

Specific comments

1. This study uses AE-33 and PAX to measure absorption. AE-33 provides the mass of absorbing aerosols as final products. Previous studies have reported the calculation of absorption coefficients from AE-33 mass concentration. However, for the sake of completeness, I would recommend to include those steps in supplementary.

**Response:** We followed the reviewer's suggestion and added the following in the revised manuscript:

"Since the AE33 aethalometer records the BC mass concentrations, the Abs(λ) at each wavelength were retrieved by getting the product of BC mass concentration ([BC]) and mass absorption cross-section (MAC) used in the instrument (Abs(λ) = [BC] × MAC) (Drinovec et al., 2015)."

2. This study used a Nafion dryer to reduce the RH of particles collected. These dryers are known to minimize particle concentration during the drying process. Is there any data on the % of particle loss within the dryer?

**Response:** We conducted a test to compare the measured light absorption coefficients

(Abs(λ)) with and without the Nafion tube. As shown in Fig. R1 below (also see Fig. S2 in the revised supporting information), the loss of Abs(λ) is little and can be ignored. We have added a sentence to show the result of this test in the revised manuscript. It reads as follows:

"As shown in Fig. S2, the loss of Abs(λ) caused by the dryer was ignored."

[Figure]

**Figure R1.** Scatter plot of light absorption coefficient measured with (Abs(λ)$_{with}$) and without (Abs(λ)$_{without}$) Nafion dryer (MD-700-24S-3). λ is the wavelength of 370, 470, 520, 590, 660, or 880 nm.

3. The Nafion dryers were connected to Aethalometers only? Aethalometer data is less susceptible to RH. But the PAX data can be influenced by high RH. Was there a dryer

connected to PAX?

**Response:** The PAX and AE33 share a same sampling tube and was set in parallel with a tee. Therefore, the PAX and AE33 were both dried by the Nafion dryer. We have clarified and added the following information in the revised manuscript:

> "It was set in parallel with the AE33 aethalometer using the same $PM_{2.5}$ cyclone and Nafion® dryer."

4. There was a $PM_{2.5}$ cyclone for Aethalometer and no cyclone for PAX. I remember the penetration efficiency of PAX reduces drastically after 1 micrometer. So, both instruments were measuring different size-cutoff particles.

**Response:** We apologize for our unclear description. As replied above, the PAX and AE33 were both collected the ambient aerosols using the same $PM_{2.5}$ cyclone. Therefore , the same size range of particles was measured by the PAX and AE33.

5. What is the area of quartz filters used?

**Response:** The area of quartz filter is 8 × 10 inch. We have added this information in the manuscript. It now reads as follows:

> "The $PM_{2.5}$ quartz-fiber filters (8 × 10 inch) (QM/A; GE Healthcare, Chicago, IL, USA) were collected during the day (from 08:00 to 20:00) and at night (from 20:00 to 08:00 the next day) using a high-volume air sampler (Tisch Environmental, Inc., USA) with a flowrate of 1.13 $m^3$ $min^{-1}$."

6. What is the flow rate of the high-volume sampler?

**Response:** The flowrate of high-volume sampler was 1.13 $m^3$ $min^{-1}$. We have added this information in the manuscript as shown above response.

7. One major shortcoming in this study is the absence of 'lensing effect' while calculating absorption. Studies have shown that the lensing effect can contribute to significant absorption. Since Aethalometer uses a filter tape to collect particles, one can

assume the core-shell structure of particles (the reason for lensing effect) gets destroyed. But the absorption from PAX will have contributions from the lensing effect. The slope of 2.29 in Figure S3 might include the lensing effect. Since the experimental setup used in this study does not measure the absorption of core-shell and core separately, it will be difficult to distinguish the contribution from the lensing effect. I would suggest the authors include this possibility in text.

**Response:** We thank the reviewer for explanation the impact of 'lensing effect' on comparison of PAX and AE33. In the revised manuscript, we have added this possible effect:

> "A slope of 2.3 was regarded as the correction factor and was comparable to the values of 2.0–2.6 reported by previous studies using a similar method (Qin et al., 2018; Tasoglou et al., 2017; Wang et al., 2019b). This difference may mainly be related to the matrix scattering and lensing effects."

8. Figure 1a – shows the apportionment of Abs, and the same is repeated as Figure 1b. Removing the repeated portion from 1a would give better visibility to it.

**Response:** We followed the reviewer's suggestion and modified this figure as shown in Fig. R2 below (also see Fig. 3 in the revised manuscript).

[Figure]

**Figure R2.** (a) Contributions of the four sources to each species from the positive matrix factorization model and (b) the light absorption of primary aerosols from each source at different wavelengths (Abs$_{pri}(\lambda)$, $\lambda$ = 370, 470, 520, 590, 660, and 880 nm)

during the study.

9. Page 2, line 13- Optical properties of LAC is not just related to its source. It also depends on the atmospheric conditions and secondary processing.

**Response:** We agree with the reviewer and revised the original sentence to:

"The optical properties of LAC aerosols are closely related to their sources as well as atmospheric conditions and secondary processing."

10. Page 4, line 3 – Educational and residential areas will have their pollution sources such as vehicles, cooking, etc.

**Response:** We agree with the reviewer. In the revised manuscript, we have revised the original description to:

"The sampling site is predominantly an educational and residential area with typical urban sources of emission including vehicles and cooking appliances."

11. Page 5, Paragraph 1 – The whole paragraph is about the analysis of filters collected. It must be specified initially.

**Response:** Following the reviewer's suggestion, we have added a sentence to clarify this in the revised manuscript:

"The collected quartz-fiber filters were used to analyse inorganic elements, carbonaceous matter, water-soluble ions, and organics."

12. Page 7, line 4 – Which PMF system was used for the analysis? I guess US EPA PMF 5.0! It needs to be mentioned with a reference.

**Response:** Yes, the version of PMF5.0 from US EPA was used in our study. We have added this information in the revised manuscript. It now reads as follows:

"The PMF version 5.0 (PMF5.0) from the US Environmental Protection Agency (Norris et al., 2014) was applied to determine the contribution of various sources

to aerosol light absorption."

13. Page 9, line 24 – Error bars on Y-axis needed. Since the X-axis is from filters (12-hour sample) and the Y-axis is the average of the same from AE-33 Abs, the error bars are required to see the spread of data.

**Response:** We followed the reviewer's suggestion, and the revised version is shown in Fig. R3 and Fig. R4 below (also see Fig. S5 and Fig. S6 in the revised supporting information).

[Figure]

**Figure R3.** Scatter plots of light absorption of black carbon at different wavelengths ($Abs_{BC}(\lambda)$, $\lambda$ = 370, 470, 520, 590, 660, and 880 nm) versus mass concentration of elemental carbon (EC). The black lines are the linear regression. The vertical error bars represent one standard deviation of $Abs_{BC}(\lambda)$.

[Figure]

**Figure R4.** Scatter plots of light absorption of brown carbon at different wavelengths ($Abs_{BrC}(\lambda)$, $\lambda$ = 370, 470, 520, 590, and 660 nm) versus mass concentration of organic carbon (OC). The black lines are the linear regression. The vertical error bars represent one standard deviation of $Abs_{BrC}(\lambda)$.

14. Page 12, line 1 – The cluster 2 back trajectory doesn't touch the Vietnam cost to influence the biomass burning. Was there a spread towards land for this cluster?

**Response:** Thanks for the reviewer pointing out this issue. In the revised manuscript, we reworked this paragraph to avoid any misunderstanding. It now reads as follows:

"Cluster #2 originated from the South China Sea near the Indochina Peninsula and accounted for 35% of the total trajectories. The $Abs_{ship}(\lambda)$ was also vital in

this cluster, accounting for 34–37% of $Abs_{pri}(\lambda)$. Fig. S10 shows that the $Abs_{pri}(\lambda)$ of Cluster #2 displayed a similar diurnal trend as that of Cluster #1. Considering that the air masses of Cluster #2 also originated from the South China Sea, the sources except for ship emissions were mainly influenced by local discharge."

Technical corrections

15. Page 2, line 26 – Don't use 'firstly'. 'First' is fine.

**Response:** Change made.

16. Page 4, line 10 – 'As described previously' – It is not described anywhere before.

**Response:** This sentence has been revised to "Afterwards, seven light emitting diodes ($\lambda$ = 370, 470, 520, 590, 660, 880, and 950 nm) in the AE33 aethalometer were used to irradiate the filter deposition spot to obtain light attenuation as previously described (Drinovec et al., 2015)."

---

## Author Comment (AC2) · 10 Sep 2020

This study performed ground measurements in a small city at the very south end of continental China along the South China Sea. The authors attempted to attribute the sources from the composition and absorption measurements using receptor models, however the analysis of the entire study is rough and the conclusions are vague for the current version.

**Response:** The authors appreciate the reviewer's valuable suggestions, and we believe that the revised manuscript has been significantly improved after considering the comments. Below are point-to-point responses.

- Many pieces of essential work are missing, which should all appear in the main figures. I only list a few examples: the time series as classified by clusters, the diurnal variations of absorption for each cluster (to exclude the possible local sources), complete statistics of all parameters are required (BC mass, AAE, PM, compositions). Please do a complete and sound analysis and just show it. Otherwise, the conclusions are based on nowhere.

**Response:** We thank the reviewer for pointing out the shortcoming of our manuscript. Following the reviewer's suggestion, we have added a Fig. R1 below (also see Fig. 1 in the revised main text) to show the time series of light absorption (Abs( $\lambda$ )) influenced by different clusters, a Fig. R2 below (also see Fig. S10 in the revised supporting information) to show the diurnal variations of Abs( $\lambda$ ) of each cluster, a Table R1 below (also see Table 1 in the revised manuscript) to summarize the optical parameters, and a Table R2 below (also see Table 2 in the revised main text) to summarize the mass concentrations of PM2.5 and chemical species.

Figure R1. Time series of hourly averaged light absorption at different wavelengths  $(Abs(\lambda), \lambda = 370, 470, 520, 590, 660, and 880 nm)$ . The different types of horizontal lines represent the four clusters of air masses.

---

## Author Response (AR2)

*Reviewer # 1*

*(1) Since Nafion dryer was used, what is the RH at the outlet of the dryer*

**Response:** The RH at the outlet of the dryer was 22 ± 7% during the campaign. We added it in the revised manuscript. It now reads as follows:

"Briefly, the collected particles were desiccated (RH = 22 ± 7%) using a Nafion® dryer (MD-700-24S-3; Perma Pure, Inc., Lakewood, NJ, USA) before measurement with the AE33 aethalometer."

*(2) Need to mention the refractive index of BC and other species mentioned in the OPAC model. Did the authors modified the refractive index in the model using the latest literature and recalculated the phase function or used the standard values?*

**Response:** In this study, we used the standard refractive index of BC and other species in the OPAC model. In the revised manuscript, we added the following:

"The standard refractive index of BC and other species in the OPAC model were used."

*Reviewer #2*

*(1) I appreciate the authors' effort in addressing most of my comments. My main concern is still that you have merged a too much fraction of local sources to the analysis, given the low time-resolution of offline analysis. The diurnal variation tended to tell the strong influence of morning and afternoon rush-hour activities, but the cluster analysis the authors relied on was not able to discriminate the other sources in a possible finer resolution. The discussion about cluster#1 seemed to be conflicting, I presume it was simply some dilution of the local pollution by the marine air mass, rather than being justified sufficiently that the "shipping emission" had been transported to the site. This also applied to the fugitive dust, you may need a higher resolution measurement to tell that period had more dust than local emission. Even for biomass burning, given it was a long transport and only lasted a short period, how could you justify you have actually*

*seen significant biomass burning than the local sources, hereby deriving any useful information regarding biomass burning?*

**Response:** We thank the reviewer's further comments about the regional transport and local emissions. As the reviewer noted, the filter-based measurements cannot further identify the exact time of the influence of morning and afternoon rush-hour activities; therefore, it is impossible to completely discriminate the local emissions. After careful consideration, we deleted the discussion about the regional transport based on the cluster analysis. Nevertheless, this behavior does not affect the subject of the study which were mainly focused on the source-specific optical properties of LAC aerosols and their radiative effects.

*(2) In addition, I'm still struggling to understand the physical meaning to combine the absorption and offline composition in one PMF analysis.*

**Response:** Similar to traditional $PM_{2.5}$ source apportionment, the chemical species in PMF were used to indicate the sources of each factor. For example, the biomass-burning factor in this study was identified by $K^+$, OC, and EC. After we obtained the contributions of each source factor to the species (e.g., chemical composition and light absorption), several optical parameters (e.g., AAE and MAC) can be retrieved from the results of optical source apportionment. One of the advantages of this approach was that it can obtain the contributions of sources to the chemical species and light absorption at the same time. Moreover, as we replied in round one response, due to the different light-absorbing aerosols having distinct absorption properties, the multiwavelength light absorption data together can better constrain the PMF results compared to those using chemical data only.

[revised manuscript text omitted]

$$Abs_{LAC}(λ) = Abs_{BC}(λ) + Abs_{BrC}(λ) = Abs(λ) - Abs_{mineral}(λ) \qquad (1)$$

where $\text{Abs}_{BC}(\lambda)$, $\text{Abs}_{BrC}(\lambda)$, and $\text{Abs}_{mineral}(\lambda)$ were absorption of light by BC, BrC, and mineral dust at $\lambda$ = 370, 470, 520, 590, 660 or 880 nm, respectively (in unit of Mm$^{-1}$). The $\text{Abs}_{mineral}(\lambda)$ was retrieved from the optical source apportionment as discussed in section 3.2. With an assumption of BC only absorbing at $\lambda$ = 880 nm, the $\text{Abs}_{BC}(\lambda)$ at wavelengths of 370, 470, 520, 590, and 660 was extrapolated as follows:

$$\text{Abs}_{BC}(\lambda) = \text{Abs}(880) \times \left(\frac{\lambda}{880}\right)^{-\text{AAE}_{BC}} \tag{2}$$

where $\text{AAE}_{BC}$ represents BC AAE, which was assumed to be 1.1 based on a study by Lack and Langridge (2013). Combining Eqs. (1) and (2) gave the following equation:

$$\text{Abs}_{BrC}(\lambda) = \text{Abs}(\lambda) - \text{Abs}(880) \times \left(\frac{\lambda}{880}\right)^{-\text{AAE}_{BC}} - \text{Abs}_{mineral}(\lambda) \tag{3}$$

From the perspective of emission and formation, the $\text{Abs}(\lambda)$ could be divided into light absorption

10 contributed by primary emissions ($\text{Abs}_{pri}(\lambda)$) and secondary formation ($\text{Abs}_{sec}(\lambda)$). Therefore, the $\text{Abs}(\lambda)$ could be calculated as follows:

$$\text{Abs}(\lambda) = \text{Abs}_{pri}(\lambda) + \text{Abs}_{sec}(\lambda) \tag{4}$$

A BC-tracer method was utilized to separate $\text{Abs}_{sec}(\lambda)$ from $\text{Abs}_{pri}(\lambda)$ and the Eq. (4) could further be developed as follows (Wang et al., 2019a):

$$\text{Abs}_{sec}(\lambda) = \text{Abs}(\lambda) - \left(\frac{\text{Abs}(\lambda)}{\text{BC}}\right)_{pri} \times [\text{BC}] \tag{5}$$

where $\left(\frac{\text{Abs}(\lambda)}{\text{BC}}\right)_{pri}$ described the ratio of $\text{Abs}(\lambda)$ to BC mass concentration in primary emissions (in unit of m$^2$ g$^{-1}$) and [BC] denoted the mass concentration of BC in the atmosphere (in unit of μg m$^{-3}$). This was retrieved from the relationship between $\text{Abs}(880)$ measured by the AE33 aethalometer and EC mass concentration. Finally, the $\left(\frac{\text{Abs}(\lambda)}{\text{BC}}\right)_{pri}$ ratio was determined using a minimum $R$-squared (MRS) method

20 previously described by Wang et al. (2019a).

**2.4 Estimation of optical parameters**

AAE reflects spectral dependence of aerosol light absorption and can be used to distinguish the chemical composition of LAC aerosols. For example, LAC aerosol dominated by BC has an AAE close to 1.0 while

the presence of BrC results in AAE larger than 1.0 (Andreae and Gelencsér, 2006). As described previously, AAE could be retrieved using a power law function as follows (Andreae and Gelencsér, 2006):

$$\text{Abs}(\lambda) = C \times \lambda^{-\text{AAE}} \tag{6}$$

where $C$ is a constant independent of wavelength.

5  Additionally, MAC could be used to reflect the light absorption capacity of aerosols. The MACs of BC and BrC at different wavelengths ($\text{MAC}_{BC}(\lambda)$ and $\text{MAC}_{BrC}(\lambda)$, respectively) were calculated with $\text{Abs}_{BC}(\lambda)$ and $\text{Abs}_{BrC}(\lambda)$ divided by the corresponding mass concentrations of BC and organic matter (OM), respectively:

$$\text{MAC}_{BC}(\lambda) = \frac{\text{Abs}_{BC}(\lambda)}{[BC]} \tag{7}$$

[revised manuscript text omitted]